# Environmental Impacts of the Brazilian Egg Industry: Life Cycle Assessment of the Battery Cage Production System

**DOI:** 10.3390/ani14060861

**Published:** 2024-03-11

**Authors:** Fabiane de Fátima Maciel, Richard Stephen Gates, Ilda de Fátima Ferreira Tinôco, Nathan Pelletier, Maro A. Ibarburu-Blanc, Natalia dos Santos Renato, Fernanda Campos de Sousa, Rafaella Resende Andrade, Guilherme Moreira de Melo Silva, Valentina Becciolini

**Affiliations:** 1Department of Agricultural Engineering, Universidade Federal de Viçosa, Viçosa 36570-900, MG, Brazil; iftinoco@ufv.br (I.d.F.F.T.); natalia.renato@ufv.br (N.d.S.R.); fernanda.sousa@ufv.br (F.C.d.S.); 2Departments of Agricultural and Biosystems Engineering and Animal Science, Iowa State University (ISU), Ames, IA 50011, USA; rsgates@iastate.edu; 3Faculties of Science (Biology) and Management, University of British Columbia, Okanagan Campus, Kelowna, BC V1V 1V7, Canada; nathan.pelletier@ubc.ca; 4Egg Industry Center, Iowa State University (ISU), Ames, IA 50011, USA; maro@iastate.edu; 5Department of Biosystems Engineering, College of Agronomy, Federal University of Goiás, Goiânia 74690-900, GO, Brazil; 6Grupo Mantiqueira Brasil, Primavera do Leste 78850-000, MT, Brazil; guilhermemoreira@mantiqueirabrasil.com.br; 7Department of Agriculture, Food, Environment and Forestry, University of Florence, 50145 Florence, Italy; valentina.becciolini@unifi.it

**Keywords:** agricultural sector, egg production, intensive system, sustainability, LCA, LCI, LCIA

## Abstract

**Simple Summary:**

Promoting sustainability in food production has become essential to meet market demands, mainly due to its tendency to expand throughout the world. Life cycle assessment is a methodology recognized for providing quantitative information on the environmental impacts caused throughout the production cycle in different categories. The objective of this study was to carry out an assessment of the life cycle, from the cradle to the farm gate, following ISO 14040 and 14044 standards, for the battery cage egg production systems and associated products in Brazil. The results showed that interventions focused on feed formulation, manure management, and the welfare of laying hens play a fundamental role in promoting sustainability in battery cage egg production systems.

**Abstract:**

Brazil stands as one of the world’s leading producers of animal protein, ranking sixth in global egg production. However, estimated growth in production demand, along with environmental impacts, represents a potential threat to the sustainability of the food system. Methods for assessing and quantifying the environmental impacts generated by Brazilian egg production remain scarce, lacking current reports on comparative effects or guiding standards. The objective of this study was to conduct a life cycle assessment from cradle to farm gate, adhering to ISO 14040 and 14044 standards, for the battery cage egg production systems and associated products in Brazil, with the aim of supporting and promoting sustainability improvements in the Brazilian egg industry. The entire life cycle modeling and process sustainability analysis were executed using the openLCA software, integrated with the Ecoinvent database. Emissions related to egg production yielded results of 65.06 kg SO_2_ eq., 27.74 kg N eq., 3086.71 kg CO_2_ eq., 75,152.66 CTUe, 2.75 × 10^−5^ CFC-11 eq., and 10,044.68 kg MJ eq. per ton of eggs produced. These findings can serve as comparative benchmarks for future studies and for analyzing data across different egg production systems in Brazil.

## 1. Introduction

Brazil is recognized as a significant player in the global animal protein market, holding prominent positions in various categories. In 2022, the country was the second-largest producer of both beef and chicken, leading the way in exporting these two products. Moreover, Brazil stands as the fourth-largest pork producer and holds the fourth position in the global pork export market [1,2]. Additionally, Brazil has claimed the sixth spot in egg production, with a staggering output exceeding 52.06 billion units [3]. According to the ABPA (Brazilian Animal Protein Association) [1], 99.6% of Brazilian egg production is destined for the national market, and only 0.44% is destined for the international market, with expectations of significant growth in egg exports in the coming years. Despite the share of the export market, where 57% consists of fresh eggs and 43% of processed ones, Brazilian eggs have found their way into families’ households in 83 consumer countries around the world. These nations are distributed across different regions, with the Middle East accounting for the largest share at 64.41% of the import market, followed by the Americas (16.42%), Asia (14.49%), Africa (1.42%), Oceania (1.37%), the European Union (1.16%), and Europe Extra-EU (0.46%) [1,3].

The anticipated increase in demand for animal-derived products, along with the corresponding environmental consequences, represents a potential threat to the sustainability of the entire food system. This encompasses production, processing, and distribution, collectively responsible for generating 21% to 37% of global greenhouse gas (GHG) emissions [4]. As reported by the Food and Agriculture Organization (FAO) [5] and the Intergovernmental Panel on Climate Change (IPCC) [6], global emissions from these systems witnessed an approximate 16% increase between 1990 and 2019, with expectations of continued growth. Therefore, it becomes imperative to identify and disseminate best practices in sustainability management with the aim of achieving reductions in production-related impacts.

Sustainability in livestock production is related to various critical aspects such as food security, public health, worker safety, biodiversity loss, economic accessibility, and animal welfare [7]. As a result, the challenge within the animal production system, especially in activities related to the egg industry, lies in a delicate balance among all these environmental factors while simultaneously meeting the existing production demands [7,8].

The prevailing egg production system in Brazil is the battery cage system, implemented in open or semi-open barns, comprising around 95% of the national egg production. Within this production system, two main types of installations are prominent, distinguished by the arrangement of the cages: the pyramid or Californian model and the vertical model [9]. On the other hand, extensive or alternative systems, including cage-free, free-range, organic, and “colonial” (traditional), collectively represent an estimated 5% of Brazilian production. However, it is noteworthy that statistical production authorities do not effectively quantify these alternative systems [9,10]. These systems represent distinct characteristics inherent to housing design and management, including feeding (particularly in the case of organic systems) and the overall cost of egg production [11].

In Brazil, there is a shortage of methods to assess and quantify the environmental impacts associated with egg production. Currently, there are no existing records for comparative effects or guidelines for enhancing production practices. As a result, the adoption of new production systems is being implemented at a slow pace due to the potential impact on the product’s pricing and its accessibility to consumers [12].

Life Cycle Assessment (LCA) is regarded as a leading tool for evaluating the environmental performance of a production system [13]. LCA is an internationally standardized method, and its application is governed by ISO14040–14044 standards [14,15]. LCA serves as a method to assess the potential environmental and human health impacts of products and services throughout their life cycle, starting from raw material extraction and covering all stages of production, transportation, manufacturing, use, and end-of-life treatment [16]. It is important to emphasize that LCA applications enable the estimation of potential environmental impacts over the life cycle of a system, quantifying, within current scientific and data limitations, the likely emissions produced and resources consumed. Furthermore, LCA identifies “hotspots” within the supply chain that may be prioritized for sustainability improvement efforts [17]. Therefore, the environmental impacts calculated through LCA should not be interpreted as absolute values but rather as relative within the scope of the study [18].

As a calculation tool, the open-source LCA software OpenLCA, version 2.0 [19], was used in conjunction with the Ecoinvent database, version 3.9.1. These databases incorporate a set of flows, including product flows, elementary flows (extracted from the environment without prior human transformation), and waste flows [20]. Processes are then established based on these flows, which are listed and quantified as inputs/outputs (through inventory data), consequently allowing the creation of a comprehensive product system. By defining a product system, it is possible to calculate the impacts generated using the desired impact assessment method [21].

LCA has been extensively applied to various egg production systems globally, including countries such as the Czech Republic [22], Canada [23,24,25,26], Mexico [27], the USA [28], Spain [29], the UK [30,31], Australia [32], the Netherlands [33], and Sweden [34], among others. Across these studies, there is a consistent identification of feed production and manure management practices as the principal contributors to the environmental impacts within the egg production chain. However, these findings collectively offer valuable insights and opportunities for interventions aimed at enhancing the sustainability of egg production.

In order to support and promote sustainability improvements in the Brazilian egg industry, this study aims to conduct an LCA from cradle to farm gate, following ISO 14044 guidelines [15], for the battery cage-based egg production system and related products. Specifically, this study seeks to quantify the use of resources and emissions attributable to the evaluated production system, including specific production inputs. It aims to identify mitigation potential and specific interventions to enhance resource use efficiency and reduce emissions from Brazilian egg production. In the future, the baseline model developed in this study could be utilized to support additional analyses and comparisons investigating strategies for environmental impact mitigation. It could also contribute to production improvement plans, such as increasing efficiency and resource use, implementing new technologies, incorporating renewable energy, optimizing natural resources, and valorizing manure, among other strategies [35,36,37,38].

## 2. Material and Methods

### 2.1. Scope of the Study

#### 2.1.1. Brazilian Egg Production Chain

This study utilized a database sourced from a farm located in the Brazilian Midwest, a region characterized by a predominantly semi-humid tropical climate, describing two well-defined seasons—dry winter and hot, rainy summer [39]. The farm is organized into five sectors: a feed mill, pullet barns, laying barns, a sorting and dispatch area, and a composting area designated for manure management.

In the upstream segment of the egg production chain, inputs are primarily sourced from the production of crops such as corn and soybeans, along with the production of one-day-old chicks [40]. These inputs are directed into the farm and specifically allocated to the feed mill and pullet barns, where they play an important role in the production of pullets.

Within the farm, a feed mill is dedicated to the production of mashed feed, and the entire manufacturing process is automated—from the measurement of inputs to the dispatch of the feed. The types of feeds produced vary in accordance with the production phases of laying hens and their specific nutritional requirements at each stage [41]. The eleven pullet barns, along with the forty-eight laying barns, are structured in the conventional system of vertically arranged cage pullet. From 1 to 14 days of age, chicks are accommodated in cages with dimensions of 0.74 × 0.58 m, hosting an average of 67 chicks per cage (64 cm^2^ per chick). Commencing on the 15th day of age, a distribution process is initiated, transferring the chicks to pullet production cages with dimensions of 0.70 × 0.80 m. These cages house an average of 17 pullets each (330 cm^2^ per pullet), and they remain in the pullet barns until the 17th week of age. After this period, there is a transfer process from the pullet barns to the laying barns, where the laying hens will stay for 88 weeks. These laying barns are equipped with cages of the same dimensions but with an average of 13 laying hens per cage (430 cm^2^ per laying hen). At the conclusion of the production cycle, after 88 weeks, spent hens are subsequently evaluated and sold to processing plants.

The eggs from the laying facilities are transferred to the grading sector through an external conveyor system. In this sector, the eggs undergo anomaly detection and are categorized based on size and quality as type A, B, or C eggs. Type A eggs, which are in perfect condition for sale, are packaged and sent to the dispatch sector. Eggs with small cracks, categorized as type B, are packaged and sent to another unit for the production of liquid eggs. The egg-breaking sector is responsible for producing whole liquid eggs, which are stored and frozen in 18-L buckets, known as industry eggs—type C. All eggs go through a series of hygiene processes before their classification, including washing, enhanced with mineral oil, and are later directed to pack and dispatch or for egg-breaking facilities.

The removal of manure from the pullet and laying barns occurs daily through automated conveyors, depositing the collected manure into external containers. These containers are then transported by truck and unloaded in the composting area, along with all the organic manure generated in the unit. Mortalities experience a dehydration process in furnaces and are also deposited in the composting area. It is worth noting that the twelve composting barns store 99% of bird manure and only 1% of other residues. Consequently, the manure undergoes a biotechnological process of organic matter decomposition under controlled aerobic conditions, and to enhance the process, low-moisture materials such as wood chips are introduced, resulting in the co-product organic compost. This compost is then sold to local producers in the region. Figure 1 illustrates the Brazilian egg production chain in a battery cage system.

#### 2.1.2. System Boundaries

The system boundaries for this study comprise all pertinent flows of materials, energy, and emissions in all considered processes, in accordance with ISO 14044 guidelines [15], ranging from cradle to farm gate. This includes all processes related to feed production (inputs, energy, and water); the production of pullets and layers of hens (chicks, feed, energy, water, and manure management); as well as all processes associated with egg grading and processing (eggs, energy, packaging, and water). The internal transportation of material flows between the stages of the supply chain is also considered. Nevertheless, it is important to note that the transportation of eggs and derivatives to retail is not considered in this study, as it occurs beyond the farm gate and falls outside the defined system boundaries.

The processing of chickens for human consumption is also excluded as it is outside the system’s boundaries. Regarding the production of organic compost, despite occurring within the farm, it is treated as a co-product derived from the reuse of manure, aligning with the criteria outlined by ISO 14044 [15], which incorporates co-product allocation criteria based on products or residues classified as reusable. In this way, the valorization of spent hens, which are no longer discarded in the environment and are now used for consumption, and the production of organic compost that directs and reuses manure as a source of organic fertilizers are both regarded as allocation factors in evaluating these two co-products. Figure 2 illustrates the system boundaries for an LCA of Brazilian egg production in a battery cage system.

#### 2.1.3. Functional Unit

Different functional units (FUs) were employed to describe the results at each stage of the production chain. In feed production, the functional unit was one ton of feed produced; in pullet production, it was 1000 pullets produced; in egg production (layers), the functional unit was one ton of eggs produced; in grading, it was one ton of graded eggs; and in liquid egg production, the functional unit was one ton of liquid eggs produced. All data are applicable to the base year of 2021.

#### 2.1.4. Co-Product Allocation

ISO 14044 [15] defines a co-product as one of two or more products originating from the same elementary process. Many agri-food production systems involve multifunctional processes, presenting a common challenge of allocating impacts among co-products [42]. In this study, the egg is considered the main product, with production categorized into Type A eggs (90.31% of production), Type B eggs (9.44%), and Type C eggs, which are marketed in refrigerated buckets (0.25%). Market estimates indicate significant variations in egg prices throughout the year 2021. According to CEPEA [43], the mean price of white eggs remained at approximately BRL127.21 per box with 360 units (equivalent to approximately 21.6 kg of eggs) in the central-west region of Brazil.

The co-products considered in this study are the valorization of spent hens and the production of organic compost derived from manure. According to information from the farm, spent hens are sent to processing plants with an average weight of 1.8 kg (ranging from 1.6 to 2 kg) and are commercialized at a rate of BRL 1.5 per kilogram of hen. Once produced, the organic compost is distributed to local producers without any stock of the finished compost. This compost has a high demand, with a market price of BRL 600 per ton. However, it is emphasized that eggs are the primary product of the system and the most significant in terms of nutrition, mass, and economic value [44].

#### 2.1.5. Cutting and Exclusion Criteria

In this study, the incubation of chicks was not considered, as the acquisition of chicks is outsourced by the farm and is outside the defined system limits. Previous studies proposed by Pelletier et al. [28], Pelletier [24,25], and Turner et al. [23] suggest that, although necessary for a comprehensive LCA, breeding batches and incubation facilities make relatively trivial contributions to the overall life cycle impacts in the egg industry.

Modeling for medication use, including antibiotics, cleaning products, and enteric fermentation for hens, as well as the maintenance of infrastructure, such as machinery and farm buildings, was excluded from the evaluation.

In this assessment, a single average value for egg mass (60 g) was considered, following the farm’s guidelines, rather than accounting for a distribution of sizes of produced eggs, as suggested by Ibarburu et al. [45].

In the present study, a single egg-producing farm was considered within the battery cage production system. However, obtaining foreground data remains a challenge in the country. It is remarkable that the evaluated farm has a high production potential, pioneeringly representing the domestic market by producing over 1.2 billion units of eggs in 2021 and experiencing growing demand in the international market.

### 2.2. Life Cycle Inventory—LCI

#### 2.2.1. Data Sources and Assumptions

The foreground system data are those directly collected from producers within the supply chain under study, incorporating inputs and outputs of products, as well as emissions related to the various stages of the production chain [24]. The foreground system data for this study were collected jointly with the farm through answers to questionnaires and meetings with the responsible sectors involved in feed production, pullet and laying hen production, egg grading, liquid egg production, and manure management, ensuring that the data source remained confidential throughout the process.

Background system data are obtained through information derived from an LCI database, with support for various types of sustainability assessments that constitute one of the most critical steps in the LCA process [46]. In this study, production and supply models for food inputs, water, energy, packaging, and transportation means were utilized, sourced from the EcoInvent database [47,48], and modified whenever possible to better align with Brazilian conditions.

##### Modeling N, P and CH_4_ Emissions from Manure

The values related to volatile solids excreted were considered in accordance with the IPCC [49,50], along with the annual quantity of excreta produced by the farm. To calculate nitrogen (N) excretion, a mass balance was performed based on feed consumption, jointly with the assessment proposed by ASAE [51] and França [52]. Phosphorus (P) excretion was also determined through a mass balance based on feed consumption. For P loss estimates, a value of 3.48% was assumed, representing the expected average losses in Brazil, in accordance with the average rates found by Rittmann et al., Piovesan, and Peles [53,54,55].

The percentages of N and P in the feed for pullets and laying hens were calculated based on the nutritional levels specified in the composition of both formulations, as provided by the farm, with the percentage of 18.5% crude protein (CP) and 0.6% phosphorus (P) for the pullet feed, and 16.7% CP and 0.52% P for the laying hen feed.

The estimates of nitrogen emissions applied to the soil were calculated by subtracting the N emitted during manure management from the nitrogen excreted. The values for nitrous oxide (N_2_O), nitrate (NO_3_), nitric oxide (NOx), ammonia (NH_3_), and methane (CH_4_) were computed using Tier 1 and 2 equations from the IPCC guidelines [49,50]. As a result, the estimated values are 4.85 kg N_2_O, 68.45 kg NO_3_, 48.92 kg NH_3_, 12.00 kg NOx, 27.01 kg CH_4_, and 3.56 kg P_2_O_5_ per 1000 units of caged pullets produced; and 1.99 kg N_2_O, 31.91 kg NO_3_, 22.42 kg NH_3_, 5.50 kg NOx, 3.18 kg CH_4_, and 1.21 kg P_2_O_5_ per ton of eggs produced. Appendix A provide a comprehensive description of the considered values.

##### Modeling of Food Inputs

The feed compositions considered comprise a total of ten formulations, with five feeds designated for the pullet phase, named Pre-Starter, Starter, Grower, Maturity, and Pre-Laying; and five feeds designated for the laying phase, named Start, Peak, Laying 1, Laying 2, and Laying 3. The Pre-Starter feed, when used, starts on the first day of the chick’s life and continues until the second week, concluding at 14 days. In most cases, the Starter feed is adopted from the first day until the sixth week, concluding at 42 days. The Grower feed is used from the 7th to the 10th week, concluding at 70 days. The Maturity feed is used from the 11th to the 15th week, ending at 105 days, and can be used up to the production phase, reaching 17 weeks. The Pre-Laying feed, when used, starts in the 16th week until the start of production in the 17th week, concluding at 119 days. For the feeds destined for the laying phase, the Start feed begins in the 18th week and continues until the 30th week, concluding at 210 days. The Peak feed, from the 31st week to the 50th week, concluded at 350 days. Laying 1 feeds from the 51st week to the 70th week, concluding at 490 days. The Laying 2 feeds from the 71st week to the 90th week, concluding at 630 days. Finally, the Laying 3 feeds from the 91st week until the disposal of the laying hen, which can vary from 770 days to 875 days when the molting process occurs. These feed compositions are estimated/used by the farm as needed. The compositions of the feeds for the pullet and laying houses can be found in Appendix A.

After analyzing the quantity of all compositions, a unified formulation was considered for both pullet and laying phases, as considered in studies proposed by Pelletier et al. [28], Pelletier [24,25], and Turner et al. [23]. In all feed compositions, corn and soy (derivatives) emerge as the predominant food ingredients. For this assessment, ingredients constituting less than 1% of the composition were not considered. The inventory models for feed production were derived from the EcoInvent database, in collaboration with the Sustell platform, which integrates major LCA agro-food databases such as Agri-footprint and GFLI [56,57,58].

#### 2.2.2. Assessment of Data Quality and Uncertainty

ISO 14044 [15] requires an assessment of data quality to ensure that low-quality data do not adversely impact the results. These assessments can contribute to improvements in data quality but also help to identify key variables for sensitivity analyses [24].

Data quality was evaluated according to each flow across all foreground processes using the standard Ecoinvent pedigree matrix, as outlined by Ciroth et al. [59], described in Appendix A. The assessment of data quality for flows in all processes was evaluated based on reliability, integrity, and temporal, geographical, and technological correlation within the modeling context. A scoring scale from 1 to 5 was employed in this system, with 1 indicating the highest quality data and 5 indicating the lowest quality data. The specific assignments for each process are presented in Appendix A [21].

Pelletier [24,25] categorizes uncertainties in LCA studies into three types: inventory data uncertainty, characterization model uncertainty, and LCI model uncertainty. Bamber et al. [60] concluded in their studies that less than 20% of LCA studies published between 2014 and 2018 reported any form of uncertainty analysis. While inventory data uncertainty is most frequently reported (82% of the studies), other sources of uncertainty are considered equally important. Monte Carlo analysis emerged as the most popular method, utilized by 61% of publications to propagate uncertainty results, regardless of the type of LCA. Using the OpenLCA software, both data quality and uncertainty values were calculated from matrices and Monte Carlo simulations [19,21].

### 2.3. Life Cycle Impact Assessment—LCIA

#### 2.3.1. Impact Assessment Method and Indicators

According to the ISO 14044 guidelines [15], the LCIA phase should consider a comprehensive set of impact categories related to the product system under study. To determine a consensus set of impact categories most relevant to egg production systems, a review of LCA studies on egg production was conducted, as outlined by Maciel et al. [17].

Among the assessed impact categories, there was significant diversity in the impact categories considered in LCA studies for eggs across different countries. The analysis of 20 sources covering 10 different countries, including the Czech Republic, Canada, the United Kingdom, Serbia, the USA, Italy, Sweden, Mexico, the Netherlands, and Spain, identified a total of 25 different impact categories. Emissions related to global warming potential (CO_2_ eq.) were unanimously considered in all studies, followed by acidification (SO_2_ eq.), eutrophication (N eq.), ecotoxicity (CTUe), ozone depletion (CFC eq.), and cumulative energy demand fossil (MJ eq.), ranging from 65% to 55% of the studies. Other related impacts were not considered, as they remained below 50% of the analyzed studies and did not align with the reality of egg production considered in Brazil. Therefore, the impact categories to be evaluated include acidification, eutrophication, global warming potential (GWP), ecotoxicity, ozone depletion, and cumulative fossil energy demand (CED), as shown in Table 1.

#### 2.3.2. Comparisons with Other Studies

##### LCIA Egg Production

International studies report the environmental performance of battery cage egg production; however, direct comparisons between studies are uncertain due to frequent differences in the considered modeling. Nevertheless, it is interesting to consider and compare the impacts reported by various studies, including Guillaume et al. [22], Turner et al. [38], Estrada-González et al. [27], Abín et al. [29], Pelletier [24,25], Pelletier et al. [28], Leinonen et al. [30,31], Wiedemann and McGahan [32], Mollenhorst et al. [33], Cederberg et al. [34], and Vergé et al. [26], to the findings of the current study.

Although there are currently no publications on LCA for Brazilian egg production, a series of LCAs have been conducted for agricultural industries in response to the growing demand for information on food products and supply chains [32]. The authors Dick et al. [61], Cardoso et al. [62], Willers et al. [63], Dick et al. [64], Leis et al. [65], Carvalho et al. [66], Barros et al. [67], Maciel et al. [68], Silva et al. [69], Lima et al. [70], Alves et al. [71], Cherubini et al. [72], and Alvarenga et al. [73] describe Brazilian LCA, considering key agro-industrial products such as beef cattle, dairy cattle, broiler chickens, pork, and broiler feed, across their various production systems.

More specific studies get into the assessment of product quality in special confinement systems, reflecting the trend toward qualitative consumption of products, as indicated by Morais et al. [74]. In their study, they concluded values of 5.03, 4.77, and 8.89 kg of CO_2_ eq./kg of live weight gain in confinement for premium, super-premium Angus, and super-premium Wagyu meats, respectively.

## 3. Results

### 3.1. Life Cycle Inventory Results

The LCI data are presented in Table 2, Table 3, Table 4, Table 5, Table 6 and Table 7, corresponding to the functional unit of each production.

The data represent a total of 171,703 tons of feed in 2021.

The data represent the production of 2,636,550 pullets in 2021.

The data represent a total egg production of 72,698.23 metric tons in the year 2021.

The data represent the total production of 65,483.60 metric tons of Grade A eggs, which are packaged and sold; 6847.14 metric tons of Grade B eggs, intended for liquid egg production; and 181.63 metric tons of Grade C eggs, which are refrigerated and sold in 18-L buckets. All production values refer to the year 2021.

The data represent the production of 1953.75 metric tons of liquid egg in the year 2021.

The collected data represent the production of 68,156 metric tons of organic compost in the year 2021.

### 3.2. Allocation Results for Co-Products

Table 8 presents the allocation of co-products, organic compost, and spent hens based on the production quantity in mass (metric tons) and their respective economic value.

The mass production percentages of eggs and organic compost are considered similar when compared (within a 3% variation) but present relevant economic values. Therefore, the economic allocation method is considered more suitable, resulting in an allocation factor of 94% for egg production, 5% for organic compost production, and 1% for the valuation of spent hens.

### 3.3. Life Cycle Impact Assessment Results

All the results of the LCIA are presented in detail in Table 9, Table 10 and Table 11, using reference units for each impact category in accordance with the egg production chain.

The higher impacts in acidifying emissions and fossil CED observed in the Grower Feed are contrasted by the Maturity Feed, which displays higher impacts in eutrophication, greenhouse gas emissions, ozone depletion, and ecotoxicity. This disparity between the two feeds is linked to the increased percentage of corn-based ingredients (1.4%) in the Grower Feed and soy derivatives (1.8%) in the composition of the Mature Feed.

The Starter Feed exhibits higher impacts related to eutrophication, greenhouse gas emissions, ozone depletion, and ecotoxicity. On the other hand, Laying Feed 2 shows higher impacts related to acidifying emissions and fossil CED. This variation between the Starter Feed and Laying Feed 2 is attributed to the increase in the percentage of soy-derived ingredients (2.2%) and meat and bone meal (1.7%) in the composition of the Starter Feed and corn (3.1%) in the composition of Laying Feed 2.

#### 3.3.1. LCIA Results—Feed Production

When analyzing the LCIA results of one ton of feed produced for pullets, according to impact categories, in acidifying emissions, corn accounts for 82.4% and soy for 17.3%; in eutrophying emissions, corn represents 51.5% and soy for 48.4%; in GHG emissions, corn accounts for 33.7% and soy for 66.1%; in ecotoxicity, corn represents 22.2% and soy for 77.8%; in ozone depletion, corn corresponds to 58% and soy to 41.7%; in fossil CED, corn corresponds to 79% and soy to 20.4% of the generated impacts. Other inputs such as bone meal, limestone, natural resources, and energy account for less than 1% of the impact generated in the production of one ton of feed for pullets.

When analyzing the LCIA results of one ton of feed produced for laying hens, in acidifying emissions, corn accounts for 87.8% and soy for 11.4%; in eutrophying emissions, corn represents 63.11% and soy for 36.5%; in GHG emissions, corn accounts for 45.05% and soy for 54.4%; in ecotoxicity, corn represents 31.6% and soy 68.2%; in ozone depletion, corn corresponds to 68.7% and soy to 30.4%; in fossil CED, corn corresponds to 85% and soy represents 13.5% of the generated impacts. Other inputs such as DDG, bone meal, limestone, natural resources, and energy account for less than 1% of the impact generated in the production of one ton of feed for laying hens.

However, for a better understanding and analysis of the impacts generated by feed production, Figure 3 illustrates the relative percentages of the impacts generated by the feeds.

It is evident that the production of feed for laying hens shows a reduction in generated impacts, ranging from 11% to 33% when compared to feed for pullets.

#### 3.3.2. LCIA Results—Pullets Production

When analyzing the LCIA results for the production of 1000 units of pullets, it is observed that manure management contributed to impact categories such as acidification, eutrophication, and greenhouse gas emissions, with relatively minor contributions to ecotoxicity, ozone depletion, and fossil CED. However, feed inputs emerge as the primary drivers for ecotoxicity, ozone depletion impacts, and fossil CED.

Considering the generated impacts of acidifying emissions, waste management accounts for 81%, while feed production contributes 18.9%. For eutrophying emissions, feed production represents 51.2%, and manure management contributes 48.7%. In greenhouse gas emissions, feed production corresponds to 82.2% and manure management to 17.3%. In terms of ecotoxicity, feed production constitutes 99.9%, with electricity contributing less than 1%. Ozone depletion is primarily driven by feed production at 98.5%, with electricity contributing 1.3%. Fossil CED is dominated by feed production at 98.1%, while electricity accounts for 1.5% of the generated impacts. Other inputs such as transportation, diesel, and water contribute less than 1% to the impact generated in the production of 1000 units of pullets. Figure 4 illustrates the percentage of impacts generated by the production of 1000 units of pullets.

#### 3.3.3. LCIA Results—Egg Production

When analyzing the LCIA results of a ton of eggs produced, it is noted that feed inputs, manure management, and pullet production (which encompass all previously reported impacts) are the main drivers of impacts associated with egg production systems. In impact categories such as ecotoxicity, ozone depletion, and fossil CED, electricity has a greater contribution than manure management due to the nature of electricity production provided by the Brazilian utility [75].

When analyzing the LCIA results, in terms of acidifying emissions, manure management accounts for 73.8%, feed production for laying hens for 13.3%, and pullet production for 12.8%. For eutrophying emissions, manure management contributes 48.6%, feed production for laying hens contributes 36.5%, and pullet production contributes 14.9%. In terms of greenhouse gas emissions, feed production for laying hens corresponds to 62.8%, manure management to 19.3%, and pullet production to 17.4%. In terms of ecotoxicity, feed production for laying hens constitutes 80.4%, and pullet production contributes 19.5%. Ozone depletion is primarily driven by feed production for laying hens at 81.5%, pullet production at 16.7%, and electricity at 1.6%. Fossil CED is dominated by feed production for laying hens at 82.5%, pullet production at 15.4%, and electricity at 1.6% of the generated impacts. Other inputs such as transportation, diesel, and water contribute less than 1% to the impacts generated in the production of one ton of eggs. Figure 5 illustrates the percentage of impacts generated by the production of one ton of eggs.

#### 3.3.4. LCIA Results—Classification Sector

When evaluating the results of the classification sector, it is possible to identify that the greatest impacts in all analyzed categories stem from egg production, consequently including the impacts of feed production, pullets, and manure management. Packaging demonstrated a higher contribution to impacts in categories related to GHG emissions (4.1%), ozone depletion (8.9%), and fossil CED (25.3%) from fossil-based plastic packaging (combs, acrylic lids, and plastic film) and cardboard boxes. Other inputs, such as electricity and water, contribute less than 1% to the impact generated in the production of one ton of classified and packaged eggs. Figure 6 illustrates the percentage of impacts generated by the production of one ton of classified eggs.

#### 3.3.5. LCIA Results—Liquid Egg Production

In all analyzed categories, the greatest impacts stem from the egg classification sector, consequently encompassing the impacts of egg production, pullets, and feed. Considering the impacts generated by acidifying emissions, egg production accounts for 99.1%. For eutrophying emissions, egg production contributes 99.2%. In terms of greenhouse gas emissions, egg production corresponds to 92.1% and transportation to 7.1%. In terms of ecotoxicity, egg production constitutes 97.4%, with transportation contributing 2.5%. Ozone depletion is primarily driven by egg production at 82.7%, with transportation at 11.8%, gas at 3.4%, and electricity at 1.9%. Fossil CED is dominated by egg production at 70.8%, with transportation at 24.8%, gas at 2.2%, and electricity at 1.7% of the generated impacts. Other inputs, such as packaging and water, contribute less than 1% to the impacts generated in the production of one ton of liquid eggs. Figure 7 illustrates the percentage of impacts generated by the production of one ton of liquid eggs.

## 4. Discussion

The production of feed is directly related to the production of pullets, laying hens (eggs), and, consequently, the generation of organic compounds through the produced manure. In the United States, as reported by Pellitier et al. [28], feed production’s primary impacts are attributed to animal co-products in feed composition, particularly those derived from ruminants. The composition of feed for pullets and laying hens in Brazil differs, with corn and soy identified as the major contributors to its environmental impact. In pullet feed production, a higher concentration of soy derivatives (27%), an elevated concentration of meat and bone meal (4%), and the non-adoption of corn-derived co-products (DDG) contribute to high values of greenhouse gas (GHG) emissions, ecotoxicity, and fossil CED. However, corn comprises 65% and 62% of the composition of pullet and laying hen feed, respectively. Through a comparison of these feeds, it becomes apparent that the incorporation of DDG in laying hen feed has already led to a reduction in the generated impacts. As part of an improvement plan, reconsidering the composition of pullet feeds to meet nutritional requirements without compromising bird development could contribute to enhancing the environmental footprints of the entire production system.

In both pullet and egg production, the most substantial impacts come from feed and manure management. The regionalized supply of feed ingredients already plays a role in enhancing sustainability performance in raw material acquisition for the farm. As part of an improvement plan in manure management, promoting the development of new formulations/compositions of feeds for the pullet and laying phases can contribute to the reduction in gas emissions from intensive animal production systems. According to Hsu, Lin, and Chiou [76], high rates of crude protein can improve productive performance but may result in a significant increase in uric acid in the animals’ blood plasma, leading to excess elimination through excreta. Therefore, incentivizing the creation of feed formulations that balance protein content to maintain productivity while minimizing environmental impacts could be a beneficial strategy.

Optimizing and managing the duration of the laying cycle can be a significant contributor to sustainability in production. In this study, variations in laying cycles were identified, ranging from 105 to 125 weeks, including the molting process. Future studies with a specific focus on determining the optimal balance between economic resources and environmental outcomes can be conducted to assess the efficiency of different laying cycles. This approach has the potential to offer valuable insights into achieving optimal sustainability in egg production systems.

The impacts generated from the grading sector are the same for egg production, with the additional consideration of energy expenses for egg grading and the use of packaging. As part of an improvement plan, measures related to energy efficiency and alternative packaging solutions should be considered to mitigate and minimize environmental impacts associated with the grading process. The adoption of more energy-efficient technologies and the exploration of eco-friendly packaging options can play an important role in promoting a more sustainable and environmentally friendly egg production and grading system.

The production of liquid eggs has emerged as a tool to re-utilize cracked or designated for disposal eggs, contributing to the reduction in manure. The market for liquid/processed eggs is experiencing growth both domestically and internationally. Despite incurring additional costs such as energy, transportation, and packaging, liquid egg production transforms what would be considered manure into a product that finds its way back to the consumer’s table. In this assessment, the most significant impacts related to liquid egg production (excluding egg production itself) are associated with the transportation of eggs to the receiving and processing unit. The receipt of cracked eggs (Type B) may need transportation from different states due to the logistics of collecting Type B eggs from various farms throughout Brazil. As part of an improvement plan, optimizing transportation logistics and exploring energy-efficient means of transportation could be considered to minimize the environmental impacts associated with this aspect of liquid egg production.

The production of organic compost serves as an alternative for reuse and environmental control within the composting area. The farm considers organic compost production as its second-largest output and sees potential in this market, especially considering the farm’s geographic location. As a result, there is an assessment of the environmental impacts generated by the composting area, which subtracts emissions resulting from the use of organic compost. This highlights that the anaerobic digestion of manure can be environmentally beneficial in other sectors of livestock farming. The overall approach aligns with sustainable practices, promoting the recycling and reuse of organic materials while minimizing environmental impacts. This commitment to sustainability reflects a holistic perspective that embraces responsible resource management and contributes to the farm’s overall environmental stewardship.

While the farm’s energy use contributes partially to the environmental impacts of the egg production life cycle, there is a broad range of technologies on the market to improve energy efficiency in production. For instance, the adoption of solar panels could significantly enhance energy efficiency compared to the use of diesel generators and wood, which, in this study, introduced uncertainties in the data due to a lack of control over the quantity of wood used and the periods of use for diesel generators. Collecting detailed data on energy sources and quantities consumed in each sector would empower producers to better understand how energy use affects the sustainability of their production. Implementing more sustainable and efficient energy practices, such as the adoption of solar panels or other renewable energy sources, can play an important role in reducing the environmental footprint of egg production. This shift toward cleaner and renewable energy sources aligns with sustainable practices and contributes to the overall environmental responsibility of the farm.

### 4.1. Result of Data Sensitivity Analysis

The choice of impact assessment method can play an important role in estimating the environmental impacts attributed to any product or system. In order to investigate the effects of this choice, additional analyses were performed using alternative LCIA methods. The calculation of GHG emissions was considered using the IPCC 2021 and CML 2016 methods, which resulted in reductions of approximately 2.4% and 3.5% in GHG emissions, respectively, as shown in Table 12.

Thus, the use of alternative LCIA methods may yield differences in the estimates of other generated impacts. Therefore, it is important to emphasize the significance of selecting and consistently applying a single impact assessment methodology for comparative purposes in future assessments.

### 4.2. Results of Data Quality and Uncertainty

At first glance, the quality of input data for most processes is generally considered high, with the exception of the evaluation of impacts associated with chick acquisition and the utilization of wood-burning generators. Nevertheless, these findings hold less significance in the context of the LCA of egg production. The data pertaining to the quantity of water, fuel, and manure generated were approximated based on the total annual information provided by the farm itself. To mitigate uncertainties in LCA models for the Brazilian egg industry, forthcoming assessments should prioritize enhancing the modeling of feed inventory, ensuring precise control over energy expenditures (energy use), and fostering the development of models to monitor and quantify resource consumption in specific sectors, such as water and fuel.

One of the primary challenges encountered is estimated to be the access limitations to data and the identified gaps in certain information during this study. Beyond the data supplied by the farm, the expansion of the national inventory database is imperative, marking a significant impediment to the advancement of LCA due to the extensive volume of necessary data. Nonetheless, ongoing research endeavors are dedicated to the adoption and dissemination of LCA in Brazil, with a specific emphasis on consolidating the methodology and establishing a comprehensive database to bolster LCA efforts.

The second major challenge encountered pertained to the modeling of emissions from manure management systems. In this analysis, a combination of IPCC Tier 1 and 2 methods, coupled with a standardized loss rate, was employed to calculate phosphorus (P) losses. While the utilization of models is generally necessary for estimating emissions in an LCA due to measurement complexities, it is important to acknowledge that the use of models can introduce uncertainties into the proposed framework. All modeling processes for nitrogen (N), phosphorus (P), and methane (CH_4_) emissions were meticulously developed and computed with the aim of directing future assessments toward the control of these emissions.

### 4.3. Result of Comparisons with Other Studies

Table 13 displays values related to the LCIA of battery cage egg production systems in various countries. In this study, the values correspond to 0.07 kg SO_2_ eq.; 3.1 kg CO_2_ eq.; 75.98 CTUe.; 2.77 × 10^−5^ kg CFC-11 eq.; and 10.07 MJ eq. per kilogram of eggs produced.

Regarding the emission results, it is evident that acidifying and eutrophying emissions exhibit minimal variations in their outcomes. Concerning ecotoxicity and ozone depletion, the current study observed an increase of 13.11 CTUe and 2.76 × 10^−5^, respectively, in comparison to the assessment conducted by Guillaume et al. [22]. Conversely, in the context of fossil CED, this study demonstrated a reduction of 6.73 in comparison to the evaluations carried out by Leinonen et al. [30,31].

The values associated with CO_2_ equivalent emissions in LCIA for egg production remain consistent across all assessments, ranging from 1.4 to 5.58 kg of CO_2_ eq. per kg of egg produced over the years 2006 to 2022. The battery cage production system in countries such as the Czech Republic, Canada, the USA, the UK, Australia, and Sweden exhibits lower carbon footprints than Brazilian egg production. In contrast, Mexico, Spain, and the Netherlands show carbon footprints larger than those of Brazilian egg production. It is noteworthy that in this study, eutrophying emissions represent a value of 0.03 kg N eq. Table 14 presents values related to the LCIA of Brazilian productions, encompassing beef cattle, dairy cattle, broiler chickens, pigs, and feed for broiler chickens across various production systems.

LCA studies for beef cattle in various production systems reveal the highest CO_2_ equivalent emissions per kilogram produced, with values ranging from 9.16 to 22.52 kg CO_2_ eq./kg live weight and 18.32 to 58.3 kg CO_2_ eq./kg carcass weight. In some LCA studies on dairy cattle, the production system may not be specified, but the focus is on the production form, such as confined, semi-confined, and pasture, with values ranging from 0.54 to 1.83 kg CO_2_ eq./kg milk. Regarding pig LCA studies, many are dedicated to evaluating different manure management systems. In the assessment proposed by Cherubini [72], the values range from 3.11 to 3.55 kg CO_2_ eq./kg live weight within the considered manure management system. In assessments related to feed production, the influence of sourcing main inputs (corn and soy) close to the feed factory is evident, varying between 0.51 and 0.75 kg CO_2_ eq./kg feed.

However, when evaluating the environmental impacts of Brazilian agro-industrial productions, egg production represents impacts relatively close to the average impacts of broiler chicken and pig production.

## 5. Conclusions

In conducting an LCA following ISO 14044, including the cradle-to-gate farm scope and cut-off and exclusion criteria for the battery cage egg production systems and related products in Brazil, environmental impacts were quantified through emissions attributable to the intensive egg production system. Emissions related to egg production were measured at 65.06 kg SO_2_ eq., 27.74 kg N eq., 3086.71 kg CO_2_ eq., 75,152.66 CTUe, 2.75 × 10^−5^ CFC-11 eq., and 10,044.68 kg MJ eq. per ton of eggs produced. When considering egg classification, emissions slightly increased to 65.78 kg SO_2_ eq., 28.26 kg N eq., 3232.93 kg CO_2_ eq., 76,676.23 CTUe, 3.04 × 10^−5^ CFC-11 eq., and 13,541.11 kg MJ eq. per ton of eggs produced and classified. Finally, the production of processed liquid eggs in another unit of the farm resulted in emissions of 85.30 kg SO_2_ eq., 36.33 kg N eq., 4355.12 kg CO_2_ eq., 100,286.33 CTUe, 4.33 × 10^−5^ CFC-11 eq., and 18,436.53 kg MJ eq. per ton of liquid eggs produced.

When comparing battery cage egg production systems in various countries, it has been observed that countries like the Czech Republic, Canada, the USA, the United Kingdom, Australia, and Sweden demonstrate lower carbon footprints compared to Brazilian production. Conversely, Mexico, Spain, and the Netherlands exhibit higher carbon footprints. However, within the broader context of evaluating the environmental impacts of Brazilian agro-industrial productions, egg production consistently stays, on average, between the impacts associated with broiler chicken and pork production.

After a comprehensive evaluation of all impacts, it becomes evident that interventions focused on feed formulation, manure management, and the welfare of laying hens play pivotal roles in promoting sustainability within the battery cage egg production system. Managing sustainability poses both challenges and opportunities for the Brazilian egg industry. However, nutrition-related interventions, management practices (considering improved animal welfare, public health, and biodiversity loss), and the adoption of new technologies in the production system hold significant promise for achieving genuine sustainability in production.

As the first LCA of the Brazilian egg industry, considering a single Brazilian farm with high production potential, the presented results can serve as comparative benchmarks for future studies and analyses of data in different egg production systems in Brazil. These findings provide a foundation for ongoing efforts to enhance sustainability practices within the industry and offer valuable insights for stakeholders seeking to implement effective interventions for a more sustainable egg production system in the country.

## Figures and Tables

**Figure 1 animals-14-00861-f001:**
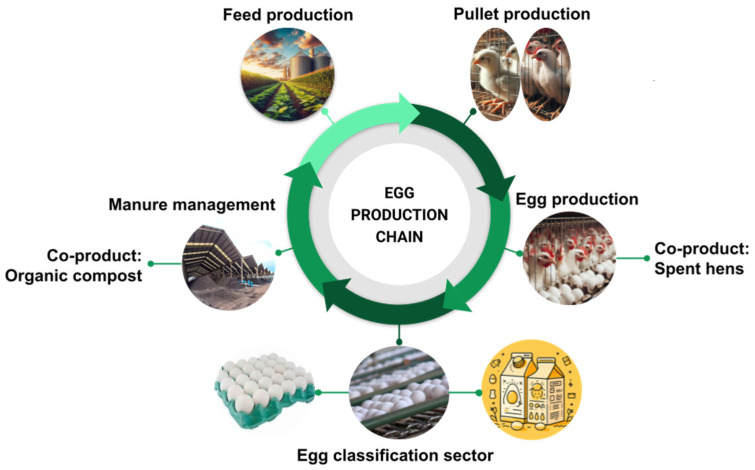
Egg production chain of Brazilian eggs in a battery cage production system. Source: The authors.

**Figure 2 animals-14-00861-f002:**
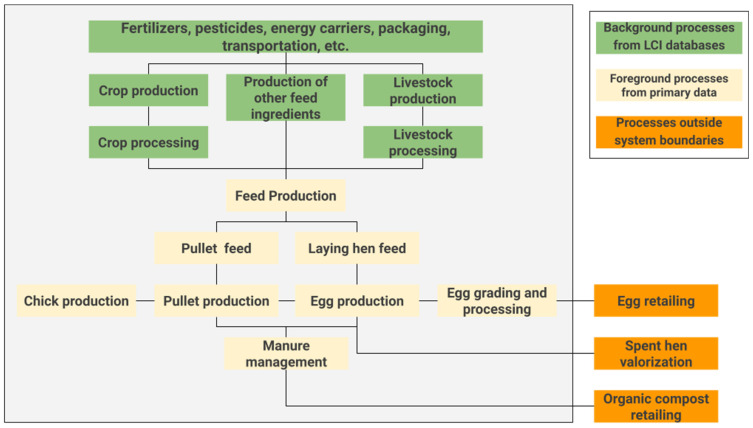
Limits of the LCA system from cradle to gate of egg production Brazilian eggs in the battery cage production system. Source: Turner et al. [23]; adapted by the authors.

**Figure 3 animals-14-00861-f003:**
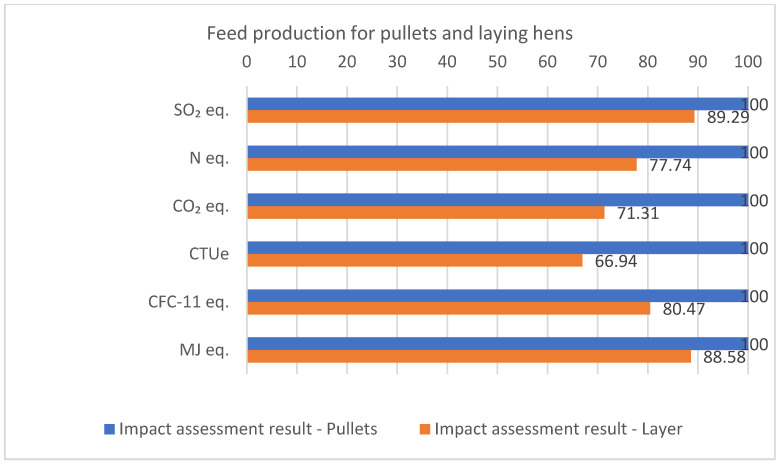
Relative percentage of environmental impacts per ton of feed produced for pullets and laying hens.

**Figure 4 animals-14-00861-f004:**
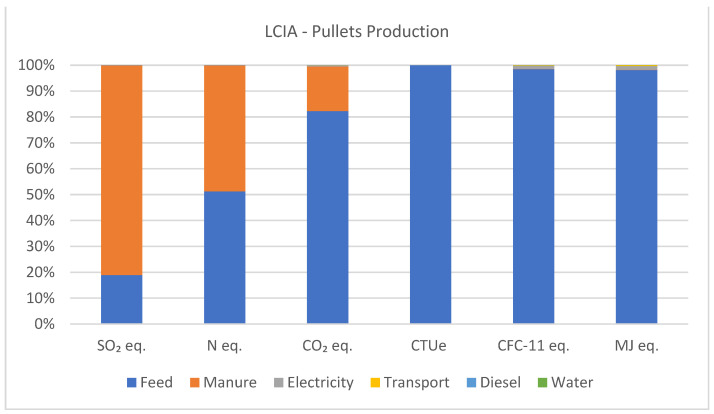
Acidifying emissions (kg SO_2_ eq.), eutrophying emissions (kg N eq.), greenhouse gas emissions (kg CO_2_ eq.), ecotoxicity (CTUe), ozone depletion (CFC-11 eq.), and fossil CED (MJ eq.) associated with the production of 1000 units of pullets produced.

**Figure 5 animals-14-00861-f005:**
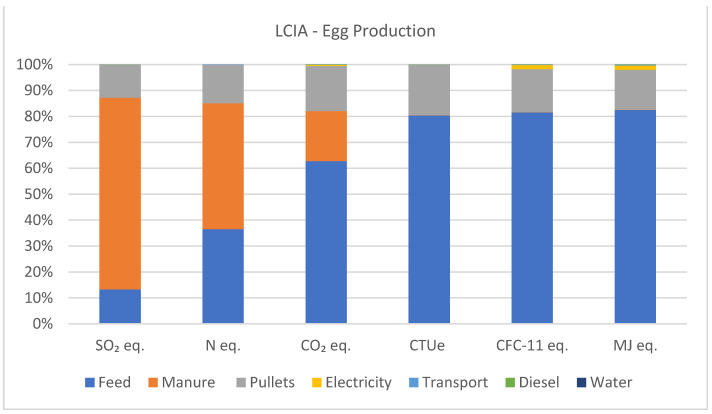
Acidifying emissions (kg SO_2_ eq.), eutrophying emissions (kg N eq.), greenhouse gas emissions (kg CO_2_ eq.), ecotoxicity (CTUe), ozone depletion (CFC-11 eq.), and fossil CED (MJ eq.) associated with the production of one ton of eggs.

**Figure 6 animals-14-00861-f006:**
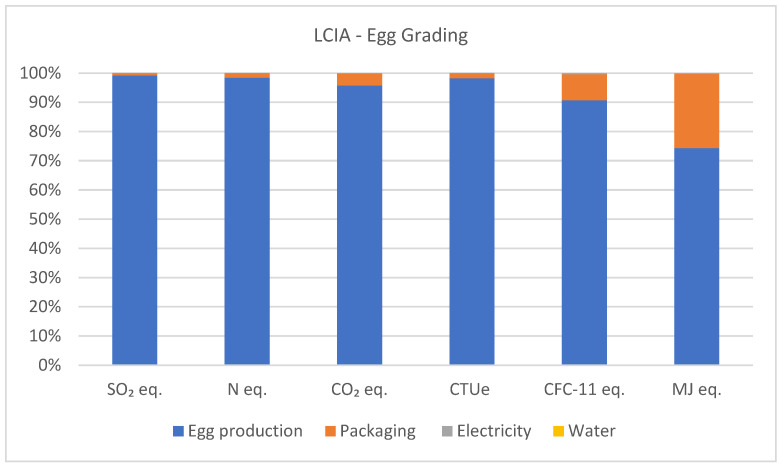
Acidifying emissions (kg SO_2_ eq.), eutrophying emissions (kg N eq.), greenhouse gas emissions (kg CO_2_ eq.), ecotoxicity (CTUe), ozone depletion (CFC-11 eq.), and fossil CED (MJ eq.) associated with the classification of one ton of eggs.

**Figure 7 animals-14-00861-f007:**
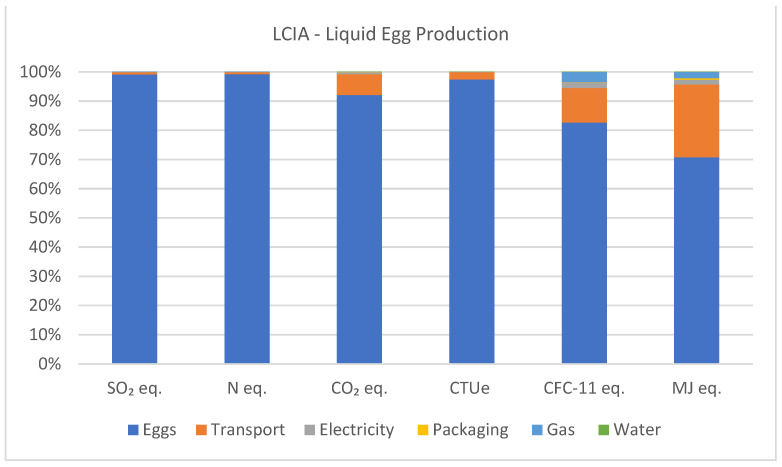
Acidifying emissions (kg SO_2_ eq.), eutrophying emissions (kg N eq.), greenhouse gas emissions (kg CO_2_ eq.), ecotoxicity (CTUe), ozone depletion (CFC-11 eq.), and fossil CED (MJ eq.) associated with the production of one ton of liquid eggs.

**Table 1 animals-14-00861-t001:** Selected impact categories for the assessment of environmental impacts in egg production and their characterization factor units.

Impact Categories	Characterization Factor Unit
Acidification	kg SO_2_ eq.
Eutrophication	kg N eq.
Global warming potential (GWP)	kg CO_2_ eq.
Ecotoxicity	CTUe (unidades tóxicas comparativas)
Ozone depletion	kg CFC-11 eq.
Cumulative Energy Demand (CED) fossil	MJ eq.

**Table 2 animals-14-00861-t002:** Percentage of feed composition for pullets and laying hens per ton of feed produced.

Ingredients	% Feed Pullets	% Feed Laying Hans
Maize	65.28	62.19
DDGs	_	8.00
Limestone	1.83	10.11
Methionine	0.29	0.19
Meat and Bone Meal	4.01	2.06
Salt	0.35	0.20
Soybean meal	20.31	14.87
Soybean hull	7.25	1.30
Vegetable oil	_	0.11
Vitamins and minerals	0.34	0.65
Soy oil	_	0.32
Others	0.34	_

**Table 3 animals-14-00861-t003:** LCI data for producing 1000 units of broiler chickens in battery cage systems.

	2021 Average Pullets Production
Inputs	
Chicks (units) ^a^	1025
Mass/Chicks (g) ^a^	35
Transportation (t*km) ^a^	37.10
Feed (tons) ^a^	5.48
Transportation (t*km) ^a^	7.31
Water (m^3^) ^ab^	14.38
Electricity (kWh) ^a^	227.32
Diesel (L) ^ab^	80.88
Outputs	
Pullets (unit)	1000
Mass (tons) ^a^	1.21
Manure (tons) ^ab^	4.27
Transportation (t*km) ^a^	5.71
N excreted (kg) ^b^	162.06
P excreted (kg) ^b^	33.29
Mortality rate (%) ^a^	2.35
Transportation (t*km) ^a^	0.02

^a^ Based on primary research data; ^b^ Data estimated, modeled, or taken from the literature; ^ab^ Total primary data and estimate (by sector).

**Table 4 animals-14-00861-t004:** LCI data for producing one ton of eggs in battery cage production systems.

	2021 Average Egg Production
Inputs	
Pullets (unit) ^a^	36.27
Distance (t*km) ^a^	11.54
Feed (tons)^a^	2.16
Distance (t*km) ^a^	2.89
Water (m^3^) ^ab^	3.07
Electricity (kWh) ^a^	102.51
Diesel (L) ^ab^	7.63
Outputs	
Eggs (tons)	1
Spent hens (kg) ^a^	51.50
Mass/spent hens (kg/hen) ^a^	1.80
Distance (t*km) ^a^	77.25
Manure (tons) ^ab^	2.06
Distance (t*km) ^a^	2.75
N excreted (kg) ^b^	57.80
P excreted (kg) ^b^	11.36
Mortality rate (%) ^a^	13.04%
Distance (t*km) ^a^	1.34

^a^ Based on primary research data; ^b^ Data estimated, modeled, or taken from the literature; ^ab^ Total primary data and estimate (by sector).

**Table 5 animals-14-00861-t005:** LCI data for producing one ton of graded eggs.

	Classified Eggs in 2021
Inputs	
Eggs (kg) ^a^	1002.56
Water (m^3^) ^ab^	4.18
Electricity (kWh) ^a^	28.82
Packaging (kg)	
Cardboard (kg) ^a^	48.36
Plastic film and acrylic (kg) ^a^	34.88
Outputs	
Eggs (Type A) (kg) ^a^	900.76
Eggs (Type B) (kg) ^a^	94.19
Eggs (Type C) (kg) ^a^	2.50
Discarded eggs (kg) ^a^	2.56

^a^ Based on primary research data; ^ab^ Total primary data and estimate (by sector).

**Table 6 animals-14-00861-t006:** LCI data for producing one ton of processed liquid egg.

	Production of Liquid Eggs in 2021
Inputs	
Eggs (kg) ^a^	1299.13
Transportation (t*km) ^a^	1295.23
Water (m^3^) ^a^	9.08
Electricity (kWh) ^a^	195.71
GLP gas (m^3^) ^a^	9.62
Packaging (kg)	1.89
Cardboard (kg)^a^	1.02
Plastic Film and Acrylic (kg) ^a^	0.87
Outputs	
Whole Liquid Eggs (kg) ^a^	1000
Eggshells (kg) ^a^	138.84
Discarded Eggs (kg) ^a^	52.63

^a^ Based on primary research data.

**Table 7 animals-14-00861-t007:** LCI data for producing one ton of organic compost.

	Production of Organic Compost in 2021
Inputs	
Manure Produced	
Poultry Manure (kg) ^ab^	2360.94
Carcasses and Other Residues (kg) ^ab^	33.09
Transportation (t*km) ^a^	28.33
Water (m^3^) ^ab^	0.08
Diesel (L) ^ab^	20.90
Shredded Eucalyptus—Wood Chips(kg) ^a^	2.78
Outputs	
Organic Compost (kg) ^a^	1000

^a^ Based on primary research data; ^ab^ Total primary data and estimate (by sector).

**Table 8 animals-14-00861-t008:** Co-product allocation and allocation percentage based on mass and economic value.

Product	Mass Production (ton.)	Economic Production (BRL)	Percentage Mass Production (%)	Percentage Economic Production (%)
Egg	72,698.23	742,026,471.7	50	94
Organic compost	68,156	40,893,600	47	5
Spent hens	3744	5,616,000	3	1

**Table 9 animals-14-00861-t009:** LCIA results per ton of feed produced for the pullet phase.

Impact Category	Pre-Starter Feed	Starter Feed	Grower Feed	Maturity Feed	Pre-Laying Feed
Acidifying emissions (kg SO_2_ eq)	4.48	4.51	4.52	4.49	4.37
Eutrophying emissions (kg N eq)	6.11	6.02	6.04	6.17	5.75
GHG emissions (kg CO_2_ eq)	1285.58	1250.66	1257.31	1303.21	1185.97
Ecotoxicity (CTUe)	42,900.81	41,391.65	41,645.30	43,595.72	39,042.66
Ozone depletion (kg CFC-11 eq)	1.31 × 10^−5^	1.29 × 10^−5^	1.29 × 10^−5^	1.32 × 10^−5^	1.24 × 10^−5^
CED fossil (MJ eq)	4332.61	4354.86	4364.24	4348.38	4214.02

**Table 10 animals-14-00861-t010:** LCIA results per ton of feed produced for the laying phase.

Impact Category	Starter Feed	Peak Feed	Laying Feed 1	Laying Feed 2	Laying Feed 3
Acidifying emissions (kg SO_2_ eq)	3.93	4.01	4.03	4.04	4.01
Eutrophying emissions (kg N eq)	4.73	4.68	4.71	4.64	4.66
GHG emissions (kg CO_2_ eq)	922.55	895.66	900.45	874.64	886.68
Ecotoxicity (CTUe)	29,169.63	27,897.89	28,045.00	26,964.00	27,534.83
Ozone depletion (kg CFC-11 eq)	1.04 × 10^−5^	1.04 × 10^−5^	1.04 × 10^−5^	1.03 × 10^−5^	1.03 × 10^−5^
CED fossil (MJ eq)	3771.96	3833.92	3854.63	3857.02	3828.50

**Table 11 animals-14-00861-t011:** LCIA results per ton of feed for pullets and laying hens, per 1000 units of pullets produced, per ton of eggs produced, per ton of graded eggs, and per ton of liquid eggs produced.

Impact Category	Feed Production (Pullets)	Feed Production (Laying Hens)	Pullets Production	Egg Production	Egg Grading	Liquid Egg Production
Acidifying emissions (kg SO_2_ eq)	4.48	4.00	129.46	65.06	65.78	85.30
Eutrophying emissions (kg N eq)	6.02	4.68	64.29	27.74	28.26	36.33
GHG emissions (kg CO_2_ eq)	1256.54	896.00	8365.25	3086.71	3232.93	4355.12
Ecotoxicity (CTUe)	41,714.92	27,922.16	228,568.90	75,152.66	76,676.23	100,286.33
Ozone depletion (kg CFC-11 eq)	1.29 × 10^−5^	1.04 × 10^−5^	7.17 × 10^−5^	2.75 × 10^−5^	3.04 × 10^−5^	4.33 × 10^−5^
CED fossil (kg MJ eq)	4322.82	3829.20	24,117.43	10,044.68	13,541.11	18,436.53

**Table 12 animals-14-00861-t012:** GHG emissions associated with the production of 1 ton of eggs using TRACI 2.1, IPCC 2021, and CML 2016 methods.

	TRACI 2.1	IPCC 2021	CML 2016
GHG emissions (kg CO_2_ eq.)	3107.90	3033.25	2999.53
Relative %	100	97.60	96.51

**Table 13 animals-14-00861-t013:** LCIA considers acidifying emissions, eutrophying emissions, GHG emissions, ecotoxicity, ozone layer depletion, and fossil CED per kilogram of eggs produced in international evaluations.

Reference	Country	Acidification (kg SO_2_ eq.)	Eutrophication (kg PO_4_ eq.)	GHG (kg CO_2_ eq.)	Ecotoxicity (CTUe)	Ozone Depletion (kg CFC-11 eq.)	CED Fossil (MJ eq.)
In this study	Brazil	0.07	_	3.1	75.98	2.77 × 10^−5^	10.07
Guillaume et al. [22]	Czech Republic	_	_	2.46	62.87	8.46 × 10^−8^	13.33
Turner et al. [23]	Canada	0.08	0.03	2.36	_	2.30 × 10^−6^	_
Estrada-González et al. [27]	Mexico	_	_	5.58	_	2.70 × 10^−7^	_
Abín et al. [29]	Spain	_	_	3.50	_	_	_
Pelletier [24,25]	Canada	0.08	0.02	2.44	_	_	11.25
Pelletier et al. [28]	USA	0.07	0.02	2.10	_	_	12.30
Leinonen et al. [30,31]	United Kingdom	0.06	0.02	2.92	_	_	16.80
Wiedemann e McGahan [32]	Australia	_	_	1.4	_	_	_
Mollenhorst et al. [33]	Netherlands	0.03	0.02	3.9	_	_	_
Cederberg et al. [34]	Sweden	_	_	1.4	_	_	_
Vergé et al. [26]	Canada	_	_	2.5	_	_	_

**Table 14 animals-14-00861-t014:** LCIA considers CO_2_ eq. emissions in Brazilian productions of beef cattle, dairy cattle, broiler chickens, pigs, and feed for broiler chickens in their different production systems per kilogram produced.

Reference	State/Region	Product	Production System	GHG Emissions (kg CO_2_ eq./kg Produced)
Dick et al. [61]	Rio Grande do Sul	Beef cattle	Extensive and Intensive	22.52 (Live weight) and 45.05 (Carcass weight) in extensive; 9.16 (Live weight) and 18.32 (Carcass weight) in intensive.
Cardoso et al. [62]	Central-west (Cerrado)	Beef cattle (in 5 different scenarios)	Extensive e Semi-intensive	58.3(C1); 40.9(C2); 29.6(C3); 32.4(C4); 29.4(C5) (All in Carcass weight).
Willers et al. [63]	Bahia	Beef cattle	Semi-intensive	9.43 (Live weight).
Dick et al. [64]	Amazonas, Cerrado, Pampa, and Pantanal	Beef cattle	Extensive	Amazonas 13.92, Cerrado 12.10, Pampa 14.62, Pantanal 21.18 (All in live weight).
Leis et al. [65]	Paraná and Santa Catarina	Dairy cattle (ECM: energy-corrected milk)	Confined, semi-confined, and pasture	0.54 confined; 0.78 semi-confined; 0.74 pasture.
Carvalho et al. [66]	Middle Southwest region of Bahia	Dairy cattle (FPCM: fat and protein corrected milk)	Semi-intensive	1.41
Barros et al. [67]	Paraná and Minas Gerais	Dairy cattle (FPCM: fat and protein corrected milk)	Confined, semi-confined	1.14 confined in PR; 1.64 semi-confined in PR; 1.83 semi-confined in MG.
Maciel et al. [68]	Minas Gerais	Dairy cattle (with and without manure treatment by anaerobic digestion)	Semi-intensive	0.88 (with treatment); 1.16 (without treatment)
Silva et al. [69]	Central west and south of Brazil	Broilers	Intensive	2.06 (Centro oeste) 1.45 (Sul)
Lima et al. [70]	Mato Grosso do Sul	Broilers	Intensive	2.70
Alves et al. [71]	Rondônia	Broilers	Intensive	3.37
Cherubini et al. [72]	_	Swine (in 4 manure management systems)	Intensive	3.50 in tanks, 3.39 in biodigestor (a), 3.11 in biodigestor (b), 3.55 in composting.
Alvarenga et al. [73]	Santa Catarina	Broiler feed	_	0.75 (CW–CW) *; 0.58 (CW–SO) *; 0.68 (SO–SW) *; 0.51 (SO–SO) *.

* CW and SO represent the central-western and southern regions of Brazil, with the acquisition of maize and soybean inputs, respectively.

## Data Availability

The data presented in this study are available upon request from the corresponding author.

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
