# Peer review of "Environmental Impacts of the Brazilian Egg Industry: Life Cycle Assessment of the Battery Cage Production System"

_animals, 2024, doi:10.3390/ani14060861_

Round 1
Reviewer 1 Report
Comments and Suggestions for Authors
Experimental design is not clearly outlined, please review accordingly. Likewise, materials and methods do not flow well in some sections. Why was fuel usage, from a chemical composition context, in transportation outside farm gates not considered? Was manure processing done at different stages of production, considering feed composition has influence on emissions. The concluding remarks should reflect on presented results and the implications, thereof - at present they seem general.

Author Response
Dear reviewer,
Please see the attachment.
Thanks again for all the corrections and contributions.
Best Regards.

Reviewer 2 Report
Comments and Suggestions for Authors
I would like to congratulate the authors for this paper that explores a very important issue. I encourage them to further discuss the concept of sustainability in terms of its implications for tightly related aspects such as public health, animal welfare, and biodiversity preservation. They do allude to food security and animal welfare, but (the implications regarding) these very important aspects warrant more explicit mentions.
Also, the paper would benefit from stressing more clearly its limitation in terms of the experimental design, and regarding how representative is this farm of the battery cage farming system in Brazil.
LL. 1-3: Please replace "Environmental Impacts of the Brazilian Egg Industry: Life Cycle Assessment of the Intensive Cage Production System" with "Environmental Impacts of the Brazilian Egg Industry: Life Cycle Assessment of the Battery Cage Production System". The concept of intensive is rather vague since it also applies to colonial/enriched cage systems, and even to some types of cage-free systems.
L. 28, L. 38, L.78, etc.: Please replace "intensive" with "battery cage" throughout the manuscript.
Ll. 73-77: Please mention other key factors related to this topic, such public health and biodiversity loss.
L. 86: Please replace “housing and conditioning of laying hens, feeding (particularly in the case of organic systems),” with “housing design and management –including feeding (particularly in the case of organic systems),”.
Ll. 89-90: Have the authors considered the case of caipira systems?
L. 238: Please stress this point again in the Conclusion section, with the goal of making clear to any reader that the full cycle assessment of this type of production practice is therefore not fully investigated in this study.
Ll. 244-246: Ditto.
Ll. 250-254: Is the farm representative of an average egg production farm in Brazil? Please discuss this point more in depth according to the evidence available, and mention it in the conclusion section.
Figures 3-7: The dark blue and dark green colors are quite difficult to differentiate when they represent small percentages (e.g., CO2 eq. In Table 7). Please change one of these colors to a more obvious different one (e.g. red).
Ll. 692-699: I would not refer to those factors as "crucial", since it is not clear whether they can significantly improve the overall results of this animal production practice in terms of sustainability. I therefore suggest using the concept of "positive role" instead.
732-736: Ditto. Please replace the term “crucial” with “important”, or a similar concept.
Ll. 757-760: Please mention more explicitly the potential for improving animal welfare, public health, and also for contributing positively to pressing environmental aspects such as biodiversity loss.
Author Response

(The authors gave the same response as above.)

Reviewer 3 Report
Comments and Suggestions for Authors
Dear authors,
the manuscript provides a detailed life cycle assessment about Intensive Cage Production System in the Brazilian Egg Industry. It is clearly written and all assumptions are well described. As there is no earlier LCA on this topic the results are really interesting an may also serve as a baseline for further studies which helps to improve the knowledge about sustainability. Please see below some suggestions to improve the article (unfortunately there were no line numbers so I was not able to address it directly to the specific line):
- You sometimes write “Life Cycle Assessment” again after you already introduced the abbreviation. For me it would be better to clearly use the abbreviation once it is defined.
- For me chapter 3.5 and 3.6 (and maybe also 3.4) are better suitable under chapter 4 (discussion) as they classify the results in the context and in relation to other studies
- In the results I would show and described the absolute values you in Table 13 in the first line as they are a highly relevant result
- Table 13: I think it would be helpful to include the year of the reference (and maybe also the housing system)
- Table 14: Please review the unit of GHG emissions, it should be also in English à kg CO2 eq./kg produced or kg CO2 per produced kg)
- Table 14: I miss the explanation of * in the last line
Kind regards
Author Response

(The authors gave the same response as above.)
